# Probabilistic Provenance Detection and Management Pathways for *Pseudotsuga menziesii* (Mirb.) Franco in Italy Using Climatic Analogues

**DOI:** 10.3390/plants10020215

**Published:** 2021-01-23

**Authors:** Maurizio Marchi, Claudia Cocozza

**Affiliations:** 1CNR—Institute of Biosciences and BioResources (IBBR), Via Madonna del Piano 10, I-50019 Sesto Fiorentino (Florence), Italy; 2Department of Science and Technology in Agriculture, Food, Environment and Forestry, University of Florence, Via S. Bonaventura 13, I-50145 Florence, Italy; claudia.cocozza@unifi.it

**Keywords:** forest management, ClimateDT, forest ecology, ecological modeling, non-native tree species, climatic normal, random forest

## Abstract

The introduction of Douglas-fir [*Pseudotsuga menziesii* (Mirb.) Franco] in Europe has been one of the most important and extensive silvicultural experiments since the 1850s. This success was mainly supported by the species’ wide genome and phenotypic plasticity even if the genetic origin of seeds used for plantations is nowadays often unknown. This is especially true for all the stands planted before the IUFRO experimentation in the 1960s. In this paper, a methodology to estimate the Douglas-fir provenances currently growing in Italy is proposed. The raw data from the last Italian National Forest Inventory were combined with literature information to obtain the current spatial distribution of the species in the country representing its successful introduction. Afterwards, a random forest classification model was run using downscaled climatic data as predictors and the classification scheme adopted in previous research studies in the Pacific North West of America. The analysis highlighted good matching between the native and the introduction range in Italy. Coastal provenances from British Columbia and the dry coast of Washington were detected as the most likely seed sources, covering 63.4% and 33.8% of the current distribution of the species in the country, respectively. Interior provenances and those from the dry coast of Oregon were also represented but limited to very few cases. The extension of the model on future scenarios predicted a gradual shift in suitable provenances with the dry coast of Oregon in the mid-term (2050s) and afterwards California (2080s) being highlighted as possible new seed sources. However, only further analysis with genetic markers and molecular methods will be able to confirm the proposed scenarios. A validation of the genotypes currently available in Italy will be mandatory as well as their regeneration processes (i.e., adaptation), which may also diverge from those occurring in the native range due to a different environmental pressure. This new information will also add important knowledge, allowing a refinement of the proposed modeling framework for a better support for forest managers.

## 1. Introduction

The arbitrary movement of non-native plant species in new environments for productive purposes or conservation strategies must be driven by scientific results [1,2]. This action may cause severe disequilibrium in the hosting ecological systems and must be planned with evidence-based information, provided by previous experiments [3]. For this reason, the introduction of new forest basic or reproductive materials is normally preceded by systematic experiments (trials), where provenances from throughout the native species range are tested across a wide range of potential planting sites and conditions [4,5].

Douglas-fir [*Pseudotsuga menziesii* (Mirb.) Franco] is a long-living tree species of the *Pinaceae* family, probably representing the most scientifically relevant and economically important North American tree species planted in Europe since the 1850s [6]. The species is naturally distributed in the western part of North America from British Columbia to California and Mexico and from Vancouver Island to the Rocky Mountains. From its native range, more than 700 populations were sampled during the 20th century and tested in the whole of Europe [5]. Known as Douglas fir, Douglas-fir, Oregon pine and Columbian pine, the huge range of Douglas-fir can be divided into two geographically distinct varieties which differ morphologically and ecologically: *P. menziesii* var. *menziesii* (coastal variety), and *P. menziesii* var. *glauca* (Rocky Mountain variety). Furthermore, within the var. *glauca*, a third genetic lineage (Mexican lineage) was identified by sequencing chloroplast DNA [7]. The high degree of genetic variability and plasticity held in its DNA sequences, which were sequenced in 2017 [8], is due to its huge natural range and is the main reason for its success across many different ecological gradients. Such characteristics allowed European countries to successfully cultivate this valuable tree from the Mediterranean to continental and Atlantic Europe [9,10].

Since it first introduction in the European environment, Douglas-fir was mainly planted in common garden experiments. The southern part of the Washington coast and Northern Oregon were often acknowledged as the most productive and suitable for the European climate [11]. The national or local experimental programs were established in many European countries, from Spain to Bulgaria and from Northern France to Southern Italy [5]. Seeds were often collected independently from the native range, mainly according to local climate and expert knowledge [12]. Therefore, many planted forests were realized using the most promising genotypes (i.e., seed sources) which were cultivated for timber and to generate new seeds for planting. In the late 1960s, the International Union of Forest Research Organizations (IUFRO) realized a continental-scale experimentation, distributing more than 180 seed sources across most of Europe. Between 1969 and 1970, both coastal and interior stands were selected in the whole native range extent [2] and according to the ecological site classification already available in the Pacific North West (PNW) of America (Figure 1). In addition to the “pure seeds” provided by IUFRO, also additional sources were sometimes tested. New seeds originated by the old growth stands already growing in the countries before the 1960s were included in some IUFRO trials. This information is only partially known but results were very promising, showing that the second-generation seeds were often able to overcome the growth performances of native ones [13]. Unfortunately, despite the interesting scientific evidence, such results remain partially obscured by the lack of knowledge concerning the genotypes used in the first “independent” and country-level experimentation of Douglas-fir [12,14]. Therefore, relating the performance in the native with that in the introduced zones is generally hard and a real genotype–environment–performance assessment has rarely been conducted [15].

Climate is known to shape the spatial distribution of forest tree species and is acknowledged as the main factor responsible for the genetic variation in forest tree species across the native range [16]. When seed sources in common garden experiments are used, variations in the expressed phenotype of a specific genotype as a function of the environmental pressures are possible, which is classified as “phenotypic plasticity” [17]. Plasticity is generally a consequence of homeostatic adjustment in another variable [18], for example, where environmental pressure alters the relations among gas exchange variables (e.g., across altitudinal gradients). The distinguishment of genetically fixed from plastic, or acclimatory characters, allows defining which plants adjust to different environments [19]. Genetic variations led to homeostasis in physiological traits of *Pseudotsuga menziesii* across environmental gradients, through isotope discrimination [20]. This was mainly attributed to genetic differences and to strong acclimatory shifts in functional traits [21,22]. The main strength of the wide experimental and reciprocal transplanting network is the ability to detect this variability and to generate large datasets for comprehensive modeling of forest tree species [23,24].

Despite the importance of Douglas-fir in Europe, its silvicultural history and distribution across a wide range of different climates, very few studies were performed, mainly due to a lack of genetic information to be coupled with direct measurements in trials and stands [12,14,25]. However, new studies have opened new research perspectives thanks to molecular data recently provided by a research group in Central Europe, where Douglas-fir old-growth stands were analyzed by means of 13 nuclear simple sequence repeats (nuSSRs) markers and classified to match the provenance origin [15,26]. Unfortunately, this information is still missing in many other European countries including Italy which is also the southernmost European country where Douglas-fir was experimented since the beginning with two IUFRO trials in Central Italy [13] and more than 20 sites already detected as old-growth stands [27]. According to the provided evidence, the lack of data concerning the genetic structure of introduced Douglas-fir stands is mandatory for a reliable assessment of available resources and future management pathways. The aim of this paper is to stimulate research around this topic and to propose a modeling framework where a probabilistic reconstruction of the most likely Douglas-fir provenances used in Italy across time and space using climatic analogues is provided. Climatic data for the PNW and Italy were here used in a regression tree classification algorithm to match the climate of origin with the conditions of the spatial locations where the species is growing in Italy. The climatic analogies were here performed for several climatic periods ranging from 1961 to 1990 (baseline climate) and until 2061–2090 to depict possible future management strategies. The goals of the proposed method are to (temporarily) supply genetic data and to build knowledge infrastructure for reliable management of the species in Italy.

## 2. Results

### 2.1. Growth Trends and Measured Trees across Italy

The data we collected from INFC and the literature depicted a wide range of dimensions and environments for Douglas-fir in Italy. The species was detected in plots ranging from 347 to 1805 m in elevation and from Mediterranean areas to high-mountain regions with cold winters, as well as hilly zones with some drought periods in summer. Douglas-fir trees were often measured in mixed stands where the species is growing with native trees such as silver fir (*Abies alba* L.), European beech (*Fagus sylvatica* L.), European black pine (*Pinus nigra* Arnold), Norway spruce (*Picea abies* H. Karst.) and some other sporadic species belonging to the *Betula* and *Sorbus* genera and very likely in Italy along the Apennines chain. Pure stands were less frequent in the dataset at around 42%. Sampled Douglas-fir trees were very variable in dimensions ranging from 5 (the minimum threshold used in the INFC2005 measurements campaign) to 79 cm in diameter at breast height (DBH) and with a total height between 4.43 and 43.77 m for an average value of 20.36 m. The database also showed large differences concerning the single tree volume, with values up to around 6 cubic meters each, which represents a good marketable tree if managed properly. Summary statistics of these and other additional mensurational parameters are reported in Table 1 and graphically shown as boxplots in Figure 2, where also the values of the native trees are reported as a unique class.

The comparison of mensurational parameters measured during the INFC2005 survey campaign showed higher values of diameter, height and total volume in Douglas-fir than in other species (Figure 2) (ANOVA: *p* < 0.05). The lack of additional tree-level data, such as the age, did not allow for a fair and extensive comparison and extensive assumptions. Age is a fundamental parameter to weight different measurements and to standardize the data. While this parameter was partially recovered from literature data for a range of ages from 5 years in the case of plots with active regeneration processes to 91 years of the oldest stand (Lagdei stand: 44.415° N, 10.018° E), deeper detail was not possible, despite the fact that the data suggest that big Douglas-fir trees are very likely in Italian forests as a result of a successful growth rate and introduction.

### 2.2. Climatic Correspondences and Provenances Detection

After tuning, the random forest classification model was fitted using 250,000 random trees using 7 variables at each split. This final setup resulted in an out of bag estimate of the error rate of 13.06%. Using the 1961–1990 climatic period (baseline), the provenances from British Columbia were estimated as the most suitable for the Italian environment and included 63.4% of the sampled plots, followed by the dry coast sources form Washington (33.8%). Only 1 location in Central Italy (Monte Amiata) was associated with provenances coming from the dry coast of Oregon and 1 with provenances coming from interior zones, which was detected in Piedmont, on the border with France in the northwest (Figure 3). The resulting distribution was mainly driven by elevation with a gradient where the interior provenances were predicted at the highest altitudinal values (close to 2000 m), followed by the plots of British Columbia and the Washington dry coast (mid-elevation around 800–1000 m). Finally, the Oregon dry coast group was located in the low-elevation locations around 450 m above sea level. According to the results, the most likely IUFRO provenances used in Italy may be allocated between IUFRO codes 1007 and 1088.

The use of the RF model on more recent climatic normal periods (1971–2000, 1981–2010 and 1991–2020) confirmed the above-mentioned provenances as the most represented and suitable even if with a decreasing trend for British Columbia, while the Washington dry cost was almost stable (Figure 4). Conversely, an increasing trend was found for the provenances of the Oregon dry coast and the provenances coming from the low-elevation sites of California. The analysis on the future scenarios (i.e., from 2011 to 2030 and until 2061–2090) showed a substantial replacement of the current groups with the Californian groups as the most important, with a stable and increasing trend. Further, other minor provenances from the interior zones were sometimes predicted by the model but with a very low use across the whole of Italy with just one or two plots occupied.

## 3. Discussion

The compiled database showed interesting productivity of the species across the Italian environment, depicting good climatic analogies between the PNW and the Italian ecological environment. Douglas-fir growth trends were often interesting and in mixed stands where competition occurs. This information is particularly important in forest ecosystems and forest management due to the need for data which include the inter- and intra-specific competition effects. Actually, this is one of the most important shortcomings of common garden experiments, where measurements are often taken until 15–20 years old, so before trees experience competition [28,29].

The proposed modeling framework and the use of future scenarios with monthly resolution showed that the climatic period between 2010 and 2050 is likely to be the most important one for the species in Italy with many provenances able to grow simultaneously. According to our analysis, the forest management strategies, which will be applied in this period, will play a key role by representing the moment when the transition between the current “genotypes” and the future one will occur. However, these predictions must be taken with care due to the lack of a real validation of the baseline matching, i.e., the provenances and the genotypes really used in the plantations, which was not possible at this time. Therefore, further research related to stress and health indicators in a climate change scenario and further ground-level counter tests are required. Furthermore, the presence and tendencies of regeneration have been highlighted as mandatory to solve the lack of important information in the National Forest Inventory dataset.

### 3.1. Climatic Analogues and Management Pathways for Italy

The ecological matching between Italy and the PNW was here the key driver for modeling. This result partially confirms previous studies [5] but is also against some others where the introduction of Douglas-fir in Europe was found to be somehow biased and often occurring in new ecological environments where the competition was artificially removed [30]. Actually, Douglas-fir is a pioneer species in the PNW, unable to generate pure stands and connected to wildfires and secondary successions [31,32]. However, the different spatial scales between our study and the previous literature available, as well as the different quality of climatic data (i.e., locally downscaled and tailored on Italy), may explain the higher climate matching that we found in this study. The provided results confirm that the northwestern part of the native range of Douglas-fir can be the main source of materials for Italy. The climates in British Columbia and the Washington coast are very close to those occurring in Italy and seem to be mainly driven by a latitudinal trend. The Italian Apennines chain is located between 45° N and 35° N and this geographic belt also matches the detected zone of the PNW. Therefore, an additional influence is contributed by altitudinal gradients to compensate the difference between a Mediterranean and an oceanic climate (i.e., coastline versus drier internal zones). This trend was also observed when future scenarios were generated with more southern ecological groups being selected and coming from California. Further, in this case, the elevation effect was found reflecting the current knowledge on climatic gradients [33,34,35]. However, we are also aware that our results include some degree of generalization; Italy is characterized by a great range of different site conditions of forest areas, with quick change in a limited spatial extent. Thus, for each region (or for each vegetational zone or ecological zone or altitudinal zone), some specific provenances (or bioclimatic group) could perform better than others. This aspect is probably the main reason to push forward the work around this valuable tree species to improve knowledge on phenotypic plasticity and adaptation at a small scale and in the Mediterranean area where the climate is also more variable than in continental Europe [36].

The use of better adapted forest tree species (genotypes) and provenance selection (genotyping) will improve the resilience of forest systems and allow assisted migration strategies [37], thus enforcing the adaptive processes of forest ecosystems [37]. High-quality climatic and molecular datasets are mandatory to support the decision process and to generate reliable predictions and modeling efforts. Wide and robust genetic datasets generated by means of dense sampling schemes and novel laboratory techniques are nowadays developed for accurate prediction considering the genotype component. Innovative and more powerful tools are often based on improved data only, not on new modeling architectures [23,24,38]. The new collection of seed sources from the native range can be an interesting option to continue the cultivation of Douglas-fir in Italy. However, this is not the only option available. Small young trees have been often measured across Italy and unpublished results and data from local projects are showing that natural regeneration processes have been observed across the whole country. Even if sometimes dangerous for non-native trees and in case of biological invasions [39,40], the occurrence of natural regeneration is always considered as a good indicator of a successful introduction for tree species in a new environment [41]. Seeds originated by existing Douglas-fir stands in Italy can be used as new potential sources that may diverge from those arriving from the PNW over long periods due to different driving forces acting on evolutionary processes and natural selection [42]. However, additional issues should be considered due to many co-occurring factors, such as a possible low genetic variability of Italian populations (i.e., founder effect) and other bottlenecks due to the used genotypes and possible biases in seed collection and nursery activities in the past. Only monitoring efforts and additional molecular data will clean the view and allow for scientifically sound comparisons.

### 3.2. Main Strength and Weakness of the Proposed Approach

The future provision of ecosystem services will be highly influenced by climate change [43] and spatial modeling techniques can support decision-makers in developing forest management strategies. Spatial modeling of forest tree species is nowadays one of the most relevant and used tools [44] where various algorithms, datasets and climatic scenarios are trained. The used algorithm has been widely tested across the whole globe but also many other modeling techniques are available and may be used. Our experience and also the available literature clearly stress the input data as the main focal point for any spatial modeling process [44,45]. Many different methods and algorithms have been used in the literature for forest tree species, such as maximum entropy models [46] and ensemble techniques [47,48]. However, only studies based on refined datasets and common garden experiments were able to fit models able to handle the genetic component such as mixed-effects models [24,49] and transfer distances and universal response functions based on parametric regressions [14,23,50], which is the target point that our study may look for in the future.

Climatic fluctuations have an important role in shaping tree species distributions and driving adaptive processes. In this framework, extreme climatic events are predicted to become more frequent and more dangerous for forest tree species in the future [51]. The resilience and resistance capacity of forest tree species is particularly demanding in ecological modeling due to the need to include them as predictors for a more robust future projection. Anyway, this information is still rare or unreliable for Douglas-fir even from large common garden experiments where tested provenances were very unlikely to experience them. This shortcoming has also probably been the main issue in previous research studies on this species where unexpected high growth rates of coastal provenances were measured also in continental environments across Europe [5,30]. As a consequence, the predictive power is dramatically reduced, which is particularly true in many forest tree species, such as Douglas-fir, where extreme frost events play a fundamental role in growth trends [12,52,53]. For this reason, the proposed modeling approach in combination with genetic data may open novel research perspectives. The new genetic data will allow the use of old-growth stands, obtained by long-term provenance trials, and the species distribution across the whole country will allow replicate measurements to cope with the lack of a spatial design. Beside the powerful method we propose, the lack of molecular data to validate our model is the main shortcoming of this modeling effort. While the tested approach demonstrates the ability of the species to grow under a wide range of climates and environments, the need remains to understand whether this ability is due to the different genotypes used, as seed sources for planting, or due to large phenotypic plasticity that is highly demanding [1,54,55] in the Italian environment.

## 4. Materials and Methods

High-quality databases are the main input data for reliable models and unbiased research results. All the data and methods used in this paper are freely available from the web as datasets or the appendix of research papers and were here just tailored and refined in order to fit the research needs and ensure robust results for the analyzed spatial extent.

### 4.1. Spatial Data Occurrence

According to the last Italian National First Inventory (INFC 2005) and available research papers published until present, Douglas-fir trees in Italy have been observed in 52 inventory plots, 20 research plots and 2 common garden experiments across the whole peninsula with a latitudinal gradient from Calabria to Piedmont (Figure 5). The abundance of the species in each plot is quite variable and ranges from 20% of the measured trees to 100% in pure stands. Thanks to the tree-level raw data provided by the NFI management team, summary statistics of the measured trees are provided in this paper and compared with the native trees growing in the same stands. In this paper, a threshold of 85% of the total standing volume was used to classify the degree of admixture [56,57]; therefore, all the stands where the standing biomass was mainly allocated on Douglas-fir trees were used for the calculation of species-specific indicators, such as the stand density index [58] and the site index [59]. A known shortcoming of INFC2005 is that the spatial coordinates of NFI plots have an uncertainty of about 1 km. This is due to privacy issues and the need to deliver the data in a freely accessible platform. However, the uncertainty in climate niche estimation forms such NFI data has been previously addressed as non-relevant for studies dealing with ecological modeling works with a country-level scale [60].

### 4.2. Climatic Data and Analogues and Scenarios

Among all the climatic datasets currently available in the literature, the CRU-TS version 4.04 [37] is the most well referenced and used in ecological studies. The database has a monthly resolution spanning from 1901 to 2019. While its primary data can be safely used at the global level, its native spatial resolution (0.5°, ~50 km at the equator) is too coarse to be reliable enough in regional studies. In order to improve the quality of CRU surfaces, two standalone software packages were used to generate downscaled climatic data for the native and the introduction range: ClimateNA [61,62] and ClimateEU [34]. ClimateNA and ClimateEU are two twin programs able to extract and downscale many climatic variables to scale-free point data for the 1901–2019 period. The core of the two systems is a high-resolution 1961–1990 baseline period (4 km for ClimateNA, 2.5 km for ClimateEU) in combination with monthly CRU-TS data to calculate historical monthly, seasonal and annual climate variables. More precisely, while the ClimateNA software was here used, an online and more recent version of ClimateEU was here exploited and available at https://ibbr.cnr.it//climate-dt/, whose name is ClimateDT.

Due to the need for evaluating the climatic shift between the native and the introduction range and considering that most of the seeds were collected across the 1960s, different 30-year climatic normal periods were used for comparison. While 1961–1990 was used to characterize the native range [5], both 1961–1990 and 1990–2019 were used for the introduction range.

The main strength of ClimateDT is that future scenarios are available with a monthly resolution until 2098. These are dynamically downscaled to scale-free locations too and in the same way as for the historical climate (CRU-TS). The future climates are derived from the global UKCP18 dataset (~60 km at the equator) which comprises results from the Met Office Hadley Centre global climate model (HadGEM3-GC3.05), as well as climate models (CMIP5) used in the latest assessment report from the Intergovernmental Panel on Climate Change [63]. UKCP18 is a global large set of climatic surfaces and uses cutting-edge climate science to provide updated observations and climate change projections. The project builds upon UKCP09 to provide the most up-to-date assessment of how the climate of the UK may change over the 21st century. In ClimateDT, this database has been mounted on ClimateEU historical series using an “anomalies” approach where the climate change forcing (anomaly) was calculated from a common normal period between CRU-TS and UKCP18 (i.e., 1961–1990 normal). Then, anomalies were added on CRU-TS to remove the intrinsic difference between the two datasets and to generate a unique and robust time series between 1901 and 2098.

The derived climatic data were processed in a statistical environment to estimate the most likely provenances for each spatial location in Italy. A random forest classification model [64] was here run in the R environment [65] to classify all the Italian stands where the species was detected. The model was trained on 758 locations available for the native range which were previously grouped in the 14 North American groups of provenances [66] and that we derived from Isaac-Renton et al. [5]. Afterwards a probability assessment was conducted in order to derive insights on the possible Douglas-fir provenances currently growing in Italy. Finally, a future management pathway scenario was built using the UKCP18 projections for 7 future climatic periods between 2020 and 2090.

## 5. Conclusions

In this paper, we have presented a modeling framework based on climatic analogies to predict the possible Douglas-fir provenances used in Italy since the 1900s and to be used as a proxy of genetic data. This information is particularly demanding due to the need for reliable models and predictions to drive future management strategies for this interesting and economically important non-native tree species. Despite the quality and reliability of the input data (National Forest Inventory and dynamically downscaled climate tailored on the target area), a large degree of uncertainty remains under our outputs. The need to confirm or reject the provided extrapolation on the used genotypes during the first planting stage is fundamental and must be addressed to validate or correct the models. The high phenotypic plasticity of forest tree species is known and only molecular markers, such as nuclear SSR or genome sequencing techniques, will be able to provide the necessary scientific robustness. In this context, our paper can be used as a potential pathway for further analysis in order to derive sampling strategies for a complete cover of the whole spatial extent where the species was planted. Additionally, UKCP18 is based on a single representative concentration pathway and a single global circulation model and more attempts are necessary to handle the model uncertainties more properly. Most of the current research has been conducted on CMIP5 projections but new pathways have been recently released, such as CMIP6. These new climatic scenarios combined with molecular data and active monitoring efforts on sampled stands, as well as new stands recruited, will improve our predictions. In this context, the research must be considered as a method to be refined when more data will be available, filling the research gap we have analyzed in this manuscript.

## Figures and Tables

**Figure 1 plants-10-00215-f001:**
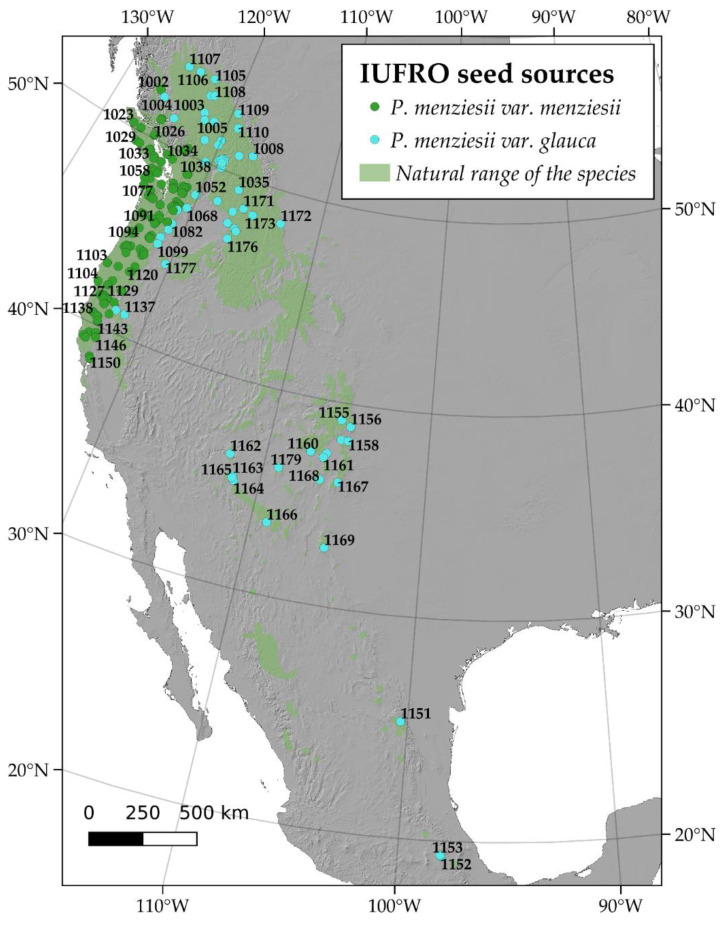
Spatial distribution of the 182 seed sources used by IUFRO for the extensive experimentation in Europe and provenances distributed to Italy and planted in the 2 IUFRO trials in Tuscany.

**Figure 2 plants-10-00215-f002:**
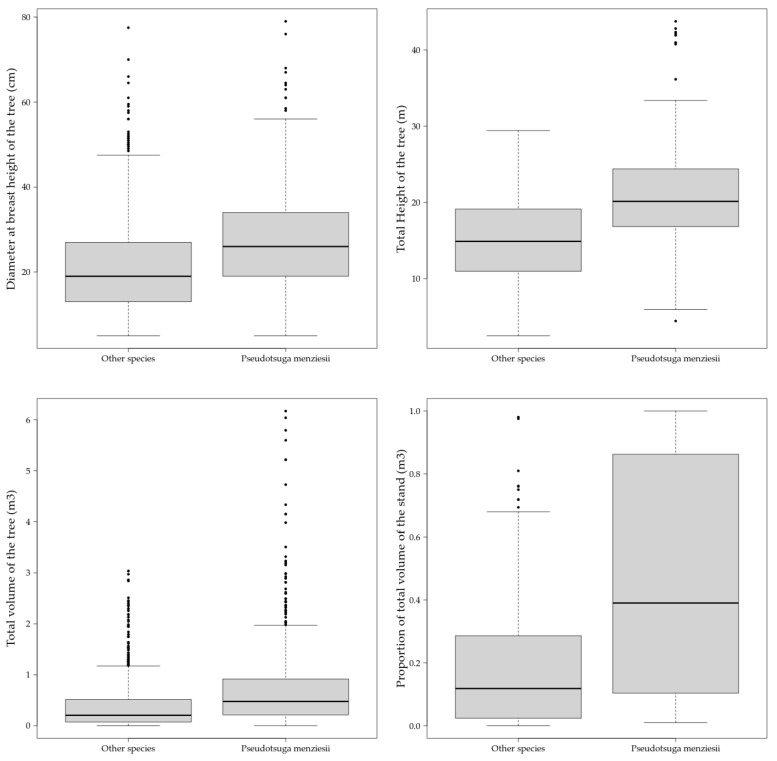
Boxplots of the main mensurational parameters measured during the INFC2005 survey campaign.

**Figure 3 plants-10-00215-f003:**
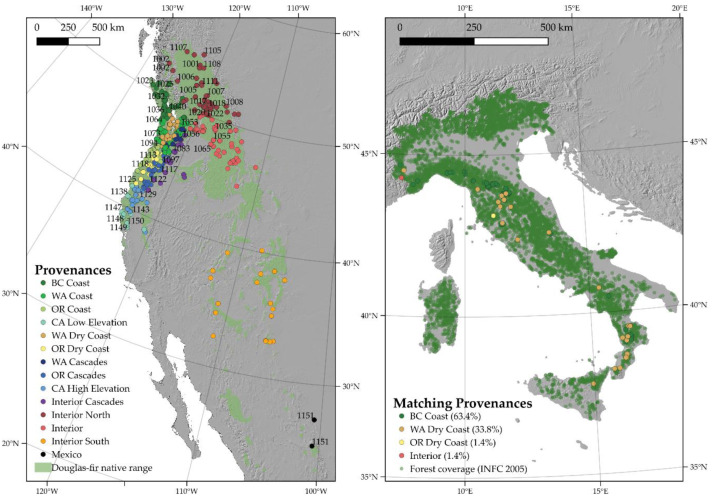
Climate matching between the PNW and the 1961–1990 normal climate in Italy and proportion of the Italia stands occupied by each bioclimatic group in the PNW. Only groups with 1 stand, at least, are reported in the legend of the Italy map.

**Figure 4 plants-10-00215-f004:**
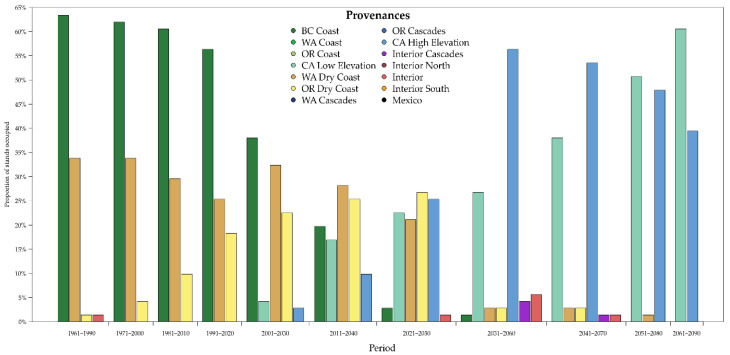
Predicted climate change effect on the proportion of Douglas-fir provenances potentially suitable in Italy in the future across the stands where this species has been detected and measured at the current time.

**Figure 5 plants-10-00215-f005:**
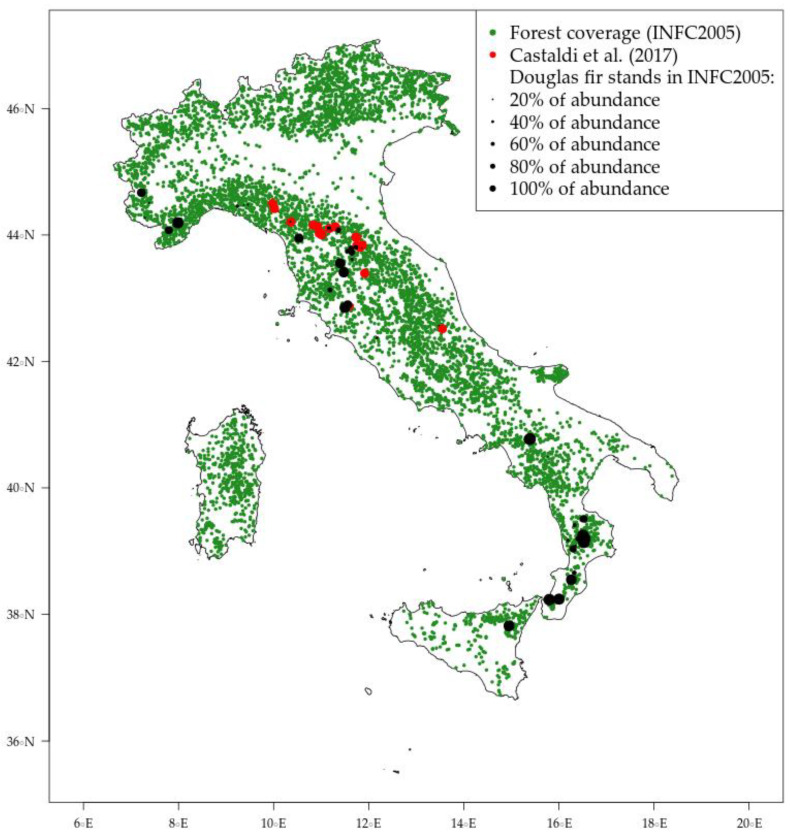
Douglas-fir distribution in Italy according to the last available National Forest Inventory and available research plots [27].

**Table 1 plants-10-00215-t001:** Summary statistics of the main mensurational parameters derived from INFC 2005 and literature data for *Pseudotsuga menziesii* in Italy. Due to the lack of information concerning the age of the trees in the National Forest Inventory dataset, this information was mainly recovered from literature data and estimated for the minimum value since the presence of small trees measured suggests successful regeneration processes.

	DBH (cm)	Height (m)	Volume (m^3^)	Curr. Vol. Inc. (m^3^·yr^−1^)	Age *	SDI	SI
Minimum	5.0	4.43	0.002	0.00008	<5 **	366.3	28.0
Average	27.24	20.36	0.732	0.02469	NA	499.3	30.9
St. dev.	11.69	5.98	0.824	0.02356	NA	291.8	4.5
Maximum	79.0	43.77	6.173	0.14032	91	837.1	36.9

* from Castaldi et al., 2017 [27] only; ** estimated from mensurational data.

## Data Availability

The spatial data concerning the distribution of Douglas-fir in North America are available from Miriam Isaac-Renton to other users upon reasonable request. The Italian dataset can be freely downloaded from the public website of the Italian National Forest Inventory (https://www.sian.it/inventarioforestale/jsp/objectives.jsp). Climatic data were generated using ClimateWNA software https://sites.ualberta.ca/~ahamann/data/climatewna.html and the ClimateDT portal https://ibbr.cnr.it//climate-dt/, both freely available from the internet.

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
