# Peer review of "Probabilistic Provenance Detection and Management Pathways for Pseudotsuga menziesii (Mirb.) Franco in Italy Using Climatic Analogues"

_plants, 2021, doi:10.3390/plants10020215_

Round 1

Reviewer 1 Report

The approach is unclear, and the structure of the manuscript is confusing. There are also some scientific wrongs in writing.

Analytically,

1) Authors report (in Abstract, lines 13-15) that “…..genetic origin of seeds used for plantations of the studied species Pseudotsuga menziesii (Mirb.) Franco is nowadays often unknown. This is especially true for all the stands planted before the IUFRO experimentation in 1960s”.

Then, they try to develop a methodology to predict the species provenances currently growing in Italy (thus, they were planted in the past).

The term predict used by authors commonly refers something to the future (in accuracy predict means=say or estimate that (a specified thing) will happen in the future or will be a consequence of something. Wrong use of terminology.

In fact, they try to “estimate” the (unknown) origin of the seeds used in Italy trials. This should be clear.

2) The introduction sections is well written.

3) What is the reason for Materials and Methods Section to follow the Results Section? This is irrational.

4) Authors write in the Results,  Line 175: “The provenances from the British Columbia were predicted as the most suitable for the Italian environment”.  

This result includes generalization; Italy is characterized by a great range of different site conditions of forest areas. Thus, for each region (or for each vegetational zone or ecological zone or altitudinal zone) some specific provenances (or bioclimatic group) could perform better than others. This should be clarified, and analytically be written.

5) Figure 2 (boxplots). Authors make a comparison between Pseudotsuga menziesii and other species. However, this includes some speculation; since the authors say (in line 156) “……This superiority was acknowledged by most of the parameters available which were very often higher than those observed for native tree species”.

Suggestions

The manuscript should be written again, following the above mentioned comments, and given special emphasis in:

1) Trying to make clear the methodological approach

2) Describing clearly the method used

3) Presenting (carefully) the results

4) Discussing correctly the findings, and soundly connecting them, with the aim of the study and those reported in the Introduction Section.

5) Make clear the weaknesses of the approach suggested, and the reason of developing it.

6) Writing, following a strict scientific way.

Author Response

Dear Reviewer, here our replies to your valiuable comments

The approach is unclear, and the structure of the manuscript is confusing. There are also some scientific wrongs in writing.

Analytically,

1) Authors report (in Abstract, lines 13-15) that “…..genetic origin of seeds used for plantations of the studied species Pseudotsuga menziesii (Mirb.) Franco is nowadays often unknown. This is especially true for all the stands planted before the IUFRO experimentation in 1960s”. Then, they try to develop a methodology to predict the species provenances currently growing in Italy (thus, they were planted in the past). The term “predict” used by authors commonly refers something to the future (in accuracy predict means=say or estimate that (a specified thing) will happen in the future or will be a consequence of something. Wrong use of terminology. In fact, they try to “estimate” the (unknown) origin of the seeds used in Italy trials. This should be clear.

Authors: the use of the verb “to predict” was initially chosen according to the “predictive Modelling approach, Finlay (2014)” stating that “Most often the event one wants to predict is in the future, but predictive modelling can be applied to any type of unknown event, regardless of when it occurred. For example, predictive models are often used to detect crimes and identify suspects, after the crime has taken place”. However, we recognize that “to predict” is more common in case of future events and that a reader may be puzzled by our paper as the Reviewer. Therefore, the suggestion to use the verb “to estimate” has been accepted through the text.

Finlay, Steven (2014). Predictive Analytics, Data Mining and Big Data. Myths, Misconceptions and Methods (1st ed.). Palgrave Macmillan. p. 237. ISBN 978-1137379276.

2) The introduction sections is well written.

Authors: thank you

3) What is the reason for Materials and Methods Section to follow the Results Section? This is irrational.

Authors: the reason relies on the Author Guidelines of the Journal. This is the structure that Plants has decided so this is a more stylistic concern related to Editorial policy.

4) Authors write in the Results, Line 175: “The provenances from the British Columbia were predicted as the most suitable for the Italian environment”. This result includes generalization; Italy is characterized by a great range of different site conditions of forest areas. Thus, for each region (or for each vegetational zone or ecological zone or altitudinal zone) some specific provenances (or bioclimatic group) could perform better than others. This should be clarified, and analytically be written.

Authors: The Reviewers is right, and we were thinking this was clear in the manuscript. We have tried to emphasize it through the text more carefully also stating that the lack of more detained data (mainly molecular data) is the main problem and that the proposed climatic approach has been already pushed as far as it could go.

5) Figure 2 (boxplots). Authors make a comparison between Pseudotsuga menziesii and other species. However, this includes some speculation; since the authors say (in line 156) “……This superiority was acknowledged by most of the parameters available which were very often higher than those observed for native tree species”.

Authors: the objection of the reviewer is pertinent, and we recognize that our comparison was unfair given the lack of information on the age and regeneration processes occurring in the analyses stands. The lack of information on this makes the comparison unreliable and we have decided to reduce the emphasis on this statement. We believe that this boxplot could be kept in the manuscript to show the dimension of Douglas-fir trees in Italy, but we warn the readers that a real comparison on growth cannot be done.

Suggestions

The manuscript should be written again, following the above-mentioned comments, and given special emphasis in:

1) Trying to make clear the methodological approach

Authors: we accepted the suggestion to improve the description of methodological approach, for example by replacing the verb “to predict” with “to estimate” and or other softer verbs.

2) Describing clearly the method used

Authors: we checked the description of methods.

3) Presenting (carefully) the results

Authors: we improved the description of the results4) Discussing correctly the findings, and soundly connecting them, with the aim of the study and those reported in the Introduction Section.

Authors: we appreciated the suggestion to reconsider the discussion, for example by reducing the emphasis on the results and on the mensurational data and mainly concerning the paragraphs which refers to Figure 2.

5) Make clear the weaknesses of the approach suggested, and the reason of developing it.

Authors: we improved the description of methodological approach. An amended version of the manuscript has been prepared following all the reviewer’s recommendations and correcting some inaccuracies and mistakes we have found, also thanks to a colleague who read the paper as well. We worked to solve the punctual issues and to include all the other suggestions We hope this new version could meet the reviewer’s ideas more properly.

6) Writing, following a strict scientific way.

Authors: we carefully checked the scientific style of the manuscript. We believe that this comment of the Reviewer was mainly referring to the structure of the paper with the M&M section after the Discussion and not between the Introduction and the Results. And this is an Editorial issue. Therefore, no changes were done on the “writing process” since we are already following the Journal’s guidelines.

Reviewer 2 Report

I would suggest highlighting the need for further research related to stress / health indicators, particularly those combined with the variability of climate change. Another point is the need for further ground level counter tests, including the presence and tendencies of regeneration. Douglas fir has shown to regenerate also in relatively dry and warm spots in sites dominated by Q. pubescens (e.i., Valtiberina, Tuscany) .

Author Response

Dear Reviewer, here our reply to your comment

I would suggest highlighting the need for further research related to stress / health indicators, particularly those combined with the variability of climate change. Another point is the need for further ground level counter tests, including the presence and tendencies of regeneration. Douglas fir has shown to regenerate also in relatively dry and warm spots in sites dominated by Q. pubescens (e.i., Valtiberina, Tuscany) .

Authors: Thank you for the positive evaluation. We have stressed the need for further research related to stress / health indicators in a climate change scenarios and the need for further ground level counter tests, including the presence and tendencies of regeneration in Discussion and Conclusions sections.

Round 2

Reviewer 1 Report

No further comments.